# Perspectives on Religious History in Early Modern Portugal: Problems, Historiographic Production and Challenges

## Paula Almeida Mendes

CITCEM, University of Porto, 4150-564 Porto, Portugal; pmendes@letras.up.pt

**Abstract:** This article aims to outline a panoramic view of the paths taken by researchers, in the last 15 years, in the field of the religious history of the Modern Period in Portugal. Starting from the identification of the scientific production and activities that have achieved a more relevant place in the framework of the study of the religious history in early modern Portugal, since the 1950s, this article draws attention to a set of subjects that urgently need to be debated. It will be argued that research in the area in question continues to constitute a challenge. The focus of this article is the European space of Portugal, not considering productions and research about Portuguese imperial spaces, namely Asia and Brazil.

**Keywords:** history of spirituality; religious history; historiography; Portugal; early modern age





## 1. Introduction

In an article published in 2009, Zulmira Santos took stock of Portuguese historiographical production on religious history and the history of spirituality in the Modern Period (Santos 2009). She outlines a double analysis, highlighting, on the one hand, the contemporaneous production of the 16th–18th centuries, and, on the other, the research developed in contemporary times around these subjects, showing that investigation polarised around these dimensions has undergone a significant evolution, especially from the middle of the 20th century. In fact, since 2009, much progress has been made in this scientific area, showing how knowledge of religious history, as well as mapping gaps and possible revisions, proves to be fundamental in the sense of recognizing religion as a structuring element of political, social and cultural life in Portugal.

Taking this framework into account, the recent production contradicts the point of view defended by Álvaro Huerga[1], who, in a chapter entitled «Portugal, terra ignota», published in the second volume of *Historia de la Espiritualidad* (Huerga 1969), emphasised the scarcity of studies on Portuguese religious history and spirituality. Even though, according to him, some fundamental works had already gone to press, such as *Historia da Igreja em Portugal* (History of the Church in Portugal) by Fortunato de Almeida (Almeida 1910–1922), *Correntes de sentimento religioso em Portugal (séculos XVI-XVIII)* (Currents of religious sentiment in Portugal (16th–18th centuries)) by José Sebastião da Silva Dias (Dias 1960), *Fr. António das Chagas: um homem e um estilo do século XVII* (Fr. António das Chagas: a man and a style of the 17th century) by Maria de Lourdes Belchior Pontes (Pontes 1953), *Vernei e a cultura do seu tempo* (Vernei and the culture of his time) by António Alberto Banha de Andrade (Andrade 1965) or the summaries by Maria de Lourdes Belchior Pontes and José Adriano de Freitas Carvalho—"Portugal (16e–18e siècles) (Portugal (16th–18th centuries))", published in the *Dictionnaire de spiritualité ascétique et mystique* (Dictionary of ascetic and mystical spirituality) (Pontes and Carvalho 1985) and *Génese e linhas de rumo da espiritualidade portuguesa* (Génese guidelines for Portuguese spirituality), which is part of the work *Antologia de Espirituais Portugueses* (Anthology of Portuguese Spirituals) (Pontes and Carvalho 1994).

It is well known that the *História Religiosa de Portugal* (Religious History of Portugal) (Azevedo 2000–2002), directed by Carlos Moreira de Azevedo, complemented by the four

volumes that make up the *Dicionário de História Religiosa de Portugal* (Dictionary of Religious History of Portugal) (Azevedo 2000–2001), made a fundamental contribution to building knowledge on ecclesiastical historiography and religious historiography in a broad sense, including literature, culture, sociability, practices and functionalities. In recent years, monographs, collective volumes of studies and editions of sources that, along with journals, such as Via *Spiritus*, *Lusitania Sacra* or *Brotéria*, or the emergence of research projects, have been developed attest to the pertinence and interest in studying the different dimensions of religious history. Taking this framework into account, we will seek to highlight some contributions that reflect the most recent research paths that have been followed since 2009, outlining a "state of the issue" in 2023. Our focus will be on the European space of Portugal, not considering the production and research about Portuguese imperial spaces, namely Asia and Brazil.

## 2. Results

Herculano Alves dedicated the third volume of *The Bible in Portugal to* the reception and ballast of Sacred Scripture in the 16th–17th centuries, analysing the impact of the Protestant Reformation on the dissemination and translation of the Sacred Scripture and its repercussions in the field of liturgy, spirituality, hagiography, sermonising, the arts, hagiography, poetry, theatre and evangelisation (Alves 2019a). In the fourth volume, *A Bíblia de João Ferreira Annes d'Almeida (1629–1690) e catálogo das suas obras bíblicas* (Alves 2019b), the author lists the national and foreign sources that allow us to reconstruct the path of the Bible translated into Portuguese by Ferreira de Almeida, outlining its historical and cultural contextualization. Chapter IV is of special interest, as it describes the various versions of the Bible translated by Ferreira de Almeida.

The publication of sources is one of the fundamental tools for building knowledge, as it allows texts to be rescued from oblivion, authorizing us to "access" the past through a "contemporary lens". In this sense, we should highlight the edition of the complete works of Father António Vieira, consisting of 30 volumes (Vieira 2013–2015), as well as texts of a devout nature, of which *Guia spiritual: breve tratado da comunhão quotidiana, e excelências da oração mental tirada dos santos* (Spiritual guide: brief treatise on daily communion, and excellences of mental prayer taken from the saints (Molinos 2017) by Miguel de Molinos is an example.

*Clavis Bibliothecarum. Catálogos e inventários* de livrarias de instituições religiosas em Portugal até 1834 (Clavis Bibliothecarum. Catalogues and inventories of libraries of religious institutions in Portugal up to 1834) by Luana Giurgevich and Henrique Leitão (Giurgevich and Leitão 2016) is an important tool for understanding the presence of written culture in monastic or convent environments. By carrying out an exhaustive survey of catalogues and inventories and covering a wide chronological span, the authors reveal a dimension that is sometimes undervalued but which is of fundamental importance for understanding the reading practices and the circulation of books in modern Portugal but also for understanding other aspects related to the "establishment" of education and literacy programmes.

The importance of books and written culture in the monastic world—which was emphasised by *Bibliografia Cronológica da Literatura de Espiritualidade, 1501–1700* (Chronological Bibliography of Literature of Spirituality, 1501–1700) (Carvalho 1988) or *Inventário da livraria de Santo António de Caminha* (Carvalho 1998) and *Inventário da livraria de Santo António de Ponte de Lima* (Carvalho 2002)—is highlighted by Fernanda Maria Guedes de Campos in her study *Para se achar facilmente o que se busca. Bibliotecas, catálogos e leitores no ambiente religioso* (So you can easily find what you're looking for. Libraries, catalogues and readers in the religious environment) (Campos 2015), which is based on her PhD thesis. Favouring an approach that looks at libraries from a structural and organisational point of view, Fernanda Campos shows that, in the monastic and conventual universe, there was an effort to organise, disseminate and (re)systematise knowledge.

A review focused on modern historiographical production would have to highlight the works that centred their attention on the Inquisition and its multiple lines of action in Portugal. The book of Ana Isabel Lopez-Salazar Codes, entitled *Inquisición y política. El gobierno del Santo Oficio en el Portugal de los Austrias (1578–1653)* (Inquisition and politics. The government of the Holy Office in Hapsburg Portugal) (Lopez-Salazar Codes 2011), which opts for an institutional approach, emphasises the internal evolution of this ecclesiastical court and the relationships of communication and collaboration, through family and clientelist networks, that are established with the Crown and other powers. On the other hand, the *História da Inquisição Portuguesa* (History of the Portuguese Inquisition) (Paiva and Marcocci 2013) by José Pedro Paiva and Giuseppe Marcocci is a fundamental contribution to our knowledge of this ecclesiastical structure, as it is "a long sedimentation of memories and controversial images, readings and interpretations that men have given, over time, of a Court that profoundly marked the past and still marks the present of the countries where it existed" (Paiva and Marcocci 2013, p. 468).

Among the various aspects of the institutional history of the Church in the early modern era that have deserved the attention of researchers is the relationship between the bishops and the Inquisition. This problem was addressed by José Pedro Paiva in *Baluartes da fé e da disciplina: o enlace entre a Inquisição e os bispos em Portugal (1536–1750)* (Bastions of faith and discipline: the link between the Inquisition and the bishops in Portugal (1536–1750)) (Paiva 2011), in which the author accurately studies the reorganization of balances of power and jurisdiction.

"The 'reforms' before the Reformation"—to use the title of a study by José Adriano de Freitas Carvalho, published in 2016 (Carvalho 2016)—encompass a topic that has received significant attention. The aforementioned work, which analyses the proposals of D. Pedro, Duke of Coimbra, with interventions carried out under the aegis of the Portuguese Crown and the various projects to renew religious life throughout the 15th century, made a fundamental contribution towards building a solid reflection on the emergence of the will of ecclesiastics and laypeople to intervene in a reformist process, inevitably imbued with an aura of renewal that predates Luther's "theses" and the atmosphere of rupture they unleashed. The focus on the Reformation was given a new lease of life with the publication of *Martinho Lutero e Portugal: diálogos, tensões e impactos* (Martin Luther and Portugal: dialogues, tensions and impacts) (Alberto et al. 2019), coordinated by Edite Martins Alberto, Ana Paula Avelar, Margarida Sá Nogueira Lalanda and Paulo Catarino Lopes, which brings together the texts of the papers presented at the International Colloquium "Martin Luther and the New Political-Religious Face of Europe", held in 2017 at the Faculty of Social and Human Sciences of the New University of Lisbon and the University of the Azores. The 500th anniversary of the posting of the "95 Theses" against Indulgences was the starting point for debate and reflection on its impact on the religious and spiritual framework of Europe, with recognised importance in redefining the political map as a result of the consequences of the wars of religion, but, also, its "export" to the world beyond the seas, where the European presence was felt.

The Council of Trent, as the point of arrival of various crossroads that had been guiding the Western scenario and the starting point for the full realisation of a set of proposals and initiatives that had been advocated, especially since the 15th century, was the subject of careful reflection in the book *O Concilio de Trento em Portugal e nas suas conquistas: olhares novos* (The Council of Trent in Portugal and its conquests: new perspectives) (Gouveia et al. 2014), coordinated by António Camões Gouveia, David Sampaio Barbosa and José Pedro Paiva. This volume, which brings together various studies presented at a series of conferences organised by the Centre for the Study of Religious History—Portuguese Catholic University (CEHR-UCP) in 2013, addresses various dimensions related to establishing a historiographical and methodological "point of situation" but also to the emergence of new perspectives, which, each in their own way, draw attention to the reception of the conciliar directives in religious life and public life and to issues which are still a subject of debate today.

The secular clergy has also received significant attention in some works, such as Hugo Ribeiro da Silva's *O Clero catedralício português e os equilíbrios sociais do poder (1564–1670)* (The Portuguese cathedral clergy and the social balances of power (1564–1670)) (Silva 2013). This study analyses the Portuguese cathedrals in post-Trent times, placing them within a framework marked by struggles and clashes between episcopal and royal powers. Moreover, local ecclesiastical institutions are also the focus of attention in *História da diocese de Viseu* (History of the diocese of Viseu) (Paiva 2016), coordinated by José Pedro Paiva, and *Dos homens e da memória: contributos para a história da diocese do Porto* (Of men and memory: contributions to the history of the diocese of Porto) (Abreu and Amaral 2018), directed by Adélio Abreu and Luís Carlos Amaral, published as part of the celebrations for the 900th anniversary of the restoration of the Diocese of Porto.

Among the various studies that have highlighted various aspects of ecclesiastical history and religious history, it is worth remembering the works that have addressed issues related to orientations on spirituality and devotions.

Gender issues in religious life were addressed in the volumes *Vozes da vida religiosa feminina. Experiências, textualidades e silêncios (séculos XV-XXI)* (Voices of Women's Religious Life. Experiences, textualities and silences (15th–21st centuries)) (Fontes et al. 2015), co-ordinated by João Luís Fontes, Maria Filomena Andrade and Tiago Pires Marques, and *Género e interioridade na vida religiosa. Conceitos, contextos e práticas* (Gender and interiority in religious life. Concepts, contexts and practices) (Fontes et al. 2017), directed by the same researchers. The first collection paved the way for a series of questions and reflections addressed by recent bibliography, such as female literary production in monastic spaces. For its part, the volume published in 2017 devotes its attention, as reflected in the 11 studies that make it up, to the contested issues surrounding religious practices that decline and develop in a kind of "interior space", which could be considered accentuated in the wake of the proposals of *Devotio Moderna*.

The writing produced by female hands, in monastic or conventual centres, was the focus of attention for Moreno Laborda Pacheco in *A mágoa do esquecimento. Escrita e memória conventual no Portugal do século XVII* (The sorrow of oblivion. Conventual writing and memory in 17th century Portugal) (Pacheco 2020). This work, which is based on the PhD thesis defended by the author in 2013, reflects the importance that writing achieved in monastic and conventual spaces, making them centres of production and conservation of culture. Moreno Laborda Pacheco also highlights that monasteries and convents were places of the exercise of "power" for the female gender, becoming clearly central to the field of women's history and literary studies. In this sense, the "Chronicles" studied by Moreno Laborda Pacheco are, in most cases, "founding writings". Luís de Sá Fardilha (Fardilha 2001, pp. 103–19) pointed out that these texts "obey" a kind of "script" that crystallises the rhetorical construction of a memory made up of different phases, which highlight the historical context in which the convent was founded and the role of its founders, the description of the geographical location and the buildings, the history and circumstances in which the life of the religious community took place, emphasising the perfect observance of the rule and the evocation, in hagiographic terms, of religious women who were "illustrious in virtue" and who died with a *reputation for sanctity*. In this sense, these texts merge edifying purposes with the need to safeguard privileges and patrimonial rights that would ensure the maintenance and survival of religious houses.

Different perspectives have focussed on the universe shaped by religious orders, offering multiple perspectives of analysis and approach. The role that religious orders and congregations have played through their historiographical eagerness to record and fix in writing the facts (both distant and present) that conformed their history (not infrequently for propaganda purposes) is well known, with a view to a strategy based on fixing and publicising their memory. This justifies the attention given to the subject in *Ao encontro* de *Histórias e Patrimónios Monásticos* (Encountering Monastic Histories and Heritage) (Marques and Osswald 2015), which brings together the contributions presented at the X Cultural Meeting of São Cristóvão de Lafões. Geraldo Coelho Dias, in a work entitled *Quando os*

*monges eram uma civilização. Beneditinos: espírito, corpo e alma* (When monks were a civilisation. Benedictines: spirit, body and soul), provides an overview of the history of the order and its influence on spirituality in Portugal (Dias 2011). The volume *Os Dominicanos em Portugal* (The Dominicans in Portugal) *(1216–2016)* (Gouveia et al. 2018), coordinated by António Camões Gouveia, José Nunes and Paulo F. de Oliveira Fontes, brings together a series of studies that analyse the presence of the Order of Preachers in Portugal, promoting a historiographical review, ranging from spirituality to architecture and art. Issue 23 of the journal of Via *Spiritus* was dedicated to the theme "The eternal in time. Memory and the construction of identities in the writing practices of religious orders" (CITCEM 2016). In turn, issue 44 of the journal *Lusitana Sacra*, published in 2021 (Fontes and Gouveia 2021), with the theme "The Capuchins of Arrábidos in a time of reform: ideology, texts and materialities", presents a series of contributions that draw attention to the activity of those religious, proven by texts, documents, libraries and other material heritage and traces.

In recent years, the military religious orders have been visibly emphasised. This is shown, for example, in Joana Lencart's study, entitled *Ordem do Templo e a Ordem de Cristo na obra de Pedro Álvares Seco no século XVI* (The Order of the Temple and the Order of Christ in the work of Pedro Álvares Seco in the 16th century) (Lencart 2022), which analyses the practice of historiography in the Order of Christ with a view to constructing an identity memory, sponsored by royal power. Contextualising the work produced by the chronicler Pedro Álvares Seco, Joana Lencart shows how the collection and selection of information and the methodology are of fundamental importance in the sense of crystallising a revisiting of the history of the order, in which, against a backdrop of nostalgia, the religious and spiritual dimension is articulated with the need to affirm and guarantee rights and privileges.

History is made through agents. Chroniclers of the various religious orders and congregations, as well as many other authors who dedicated themselves to writing the "lives" of saints and devout "lives", could not, quite naturally, fail to be aware of the centrality of the "virtues" and, not infrequently, miracles, of their members, emulating their high examples in authentic behavioural guidelines, proposed for imitation by the faithful and undoubtedly more fascinating and attractive than the texts of a more normative nature, such as the Rules and Constitutions, "stripped" of such valued dimensions as heroism and the "marvellous", which the latter used almost to the point of exhaustion. The works of Paula Almeida Mendes (Mendes 2017), José Félix Duque (Duque 2016) and Leonardo Coutinho de Carvalho Rangel (Rangel 2018) analyse the centrality of "holiness" as a factor of prestige and distinction in Catholic territories in post-Trent times. Supported in various sources, the authors of these texts tried to seduce various and diverse audiences in order to convince them that Christian perfection, which would ensure eternal salvation, was something accessible to all the faithful and possible in all states. The sources used show a wide variety of cases and a rhetorical discourse aimed at persuading readers to reform and correct habits of their behaviour and solidify their spiritual and devout practices or the Holy See to officially recognise the exceptionality of the biographers.

The issue surrounding relics has attracted the attention of several researchers recently. It is well known that the continuous accumulation of relics (not infrequently fake) in the West, especially throughout the Middle Ages and, above all, the reformist polemic surrounding their veneration, contested by Erasmus and Calvin, logically provoked a reaction at the Council of Trent, which reactivated their cult through the decree *De invocatione, veneratione, et reliquis sanctorum,* approved on 3 and 4 December 1563, in session XXV. In fact, the years of affirmation and consolidation of the decisions of Trent revalorised spiritual and devotional practices of greater external expressiveness, especially in terms of collective devotions (e.g., processions), the cult of saints, images and relics. Rosa Capelão's work, entitled *El culto de reliquias en Portugal en los siglos XVI y XVII: contexto, norma, función y simbolismo* (The cult of relics in Portugal in the 16th and 17th centuries: context, norm, function and symbolism) (Capelão 2022), made an important contribution towards recognising the relevance of studying this dimension, which could be considered amplified thanks to the exhibition catalogue *Reliquias? O projeto Reliquiarum*" (Relics? The Reliquiarum Project),

coordinated by António Camões Gouveia (Montenegro 2022a), and the volume of studies entitled *Relíquias/Relics* (Montenegro 2022b), which raises a series of issues ranging from the functionalities of relics to their typologies, their impact and the links they establish with devotees. These studies were accompanied by exhibitions designed to communicate these realities to wider audiences.

In turn, the "eighteenth-century inventory" of miraculous Marian images, i.e., the *Santuário Mariano* (Marian Sanctuary) of Fr Agostinho de Santa Maria, was the subject of a study by Maria de Lurdes Correia Fernandes, entitled "Imagens e devoções marianas na época moderna: o testemunho do *Santuário Mariano* (1707–1723) de Fr. Agostinho de Santa Maria, OSA" (Marian images and devotions in modern times: the testimony of the Marian Sanctuary (1707–1723) of Fr Agostinho de Santa Maria, OSA), included in the volume *De Cister a outros espaços e caminhos: as Beiras e as suas expressões histórico-culturais* (From Cistercian to other spaces and paths: Beiras and its historical-cultural expressions), which brings together the works presented at the XII Cultural Meeting of Lafões (Fernandes 2017, pp. 47–72). Maria de Lurdes Correia Fernandes inserts the work into the abundant bibliographical production of the 16th–17th centuries, emphasising the specific character of this monumental work produced by this Augustinian hermit.

The issue of court spirituality, which had already been discussed in the volume *Espiritualidade e Corte em Portugal (séculos XVI a XVIII)* (Faculdade de Letras da Universidade do Porto 1993), was given a new lease of life in the various articles published in issue 27 of the *Archivio Italiano per la Storia della Pietà*, on the theme "Percorsi di spiritualità alla corte portughese in età Moderna/Caminhos de espiritualidade da Corte portuguesa na Época Moderna" (Paths of spirituality of the Portuguese Court in the Modern Period) (Carvalho and Santos 2014), coordinated by José Adriano de Freitas Carvalho and Zulmira Coelho Santos. The works collected in this volume study manifestations of religious sentiment and the ways in which they are shaped into representations and practices, ranging from devotions and readings to networks of solidarity between lay people and the clergy.

The circulation of models of devout behaviour, particularly within the framework of spiritual practices and readings in female aristocratic circles, was the subject of Ana Cecília Costa's attention in her PhD thesis *S. Francisco de Sales em Portugal. Elementos bibliográficos para o estudo da sua Obra e espiritualidade* (St. Francis of Sales in Portugal. Bibliographical elements for the study of his work and spirituality St Francis de Sales in Portugal) (Costa 2014), which analyses the reception of Salesian spirituality in the Portuguese context.

Studies on the Jews or Jewish spirituality have received significant attention within scientific production. This is shown in *O Porto judaico: encruzilhadas de vidas nos caminhos da história* (The Jewish Porto: crossroads of lives on the paths of history) (Mea 2020) by Elvira Mea, or the volume coordinated by Maria Marta Lobo de Araújo, Maria de Fátima Reis and Bernardo José Ferreira Reis: *Caridade e assistência na diáspora sefardita (séculos XVI-XVIII). Contributos documentais* (Charity and assistance in the Sephardic diaspora (16th–18th centuries. Documentary contributions) (Reis et al. 2019).

### 3. Conclusions

Despite the contributions mentioned above, which clearly indicate the relevance and urgency of studying the multiple dimensions presented, there is still a long and complex research itinerary to be developed, shaped by various challenges.

Zulmira Santos, in an accurate and pertinent way, listed several gaps, as the absence of "a complete roadmap of spirituality in Portugal, despite the *Bibliografia Cronológica da Literatura de Espiritualidade. 1501–1700* (Chronological Bibliography of Literature of Spirituality. 1501–1700), published by the Faculty of Letters of the University of Porto in 1988, because it ends in the 17th century" (Santos 2009, p. 257). She also emphasized the lack of studies about the "development of a catechetical and moralising literature" (Santos 2009, p. 258), the prophetic currents (Santos 2009, p. 260), the "Catholic Illustration" or the Jacobean (Santos 2009, pp. 260–61). Although, as we have tried to show, important contributions have been made in the last decade, there are still dimensions that raise questions and doubts.

One of these aspects concerns lay people who, longing for new forms of Christian/spiritual life without renouncing their state, want to escape a "routine" religious practice by investing in certain devout practices. Relations between these laypeople and the clergy have not always been peaceful, as is well known. Public and professional life, as well as married life, appear as obstacles that condition the progress of the spiritual life and an itinerary aimed at achieving perfection, seen through the prism of the cloister. In this sense, it would be worthwhile once again to consider the background to the *modern Devotio* in Portugal, without forgetting, of course, the complexity of this issue. In fact, the search for a more affective relationship with God and more "effective" ways of communicating with the Divine took on a wide variety of representations and modalities, which, above all, proposed methods to help pray outside the space of the liturgy and the divine office but which also stimulated spiritual practices of various kinds. This "openness" brought about by the emergence of the *Devotio Moderna* was, in fact, a fundamental way forward for many laypeople who wanted to participate more actively and visibly in the field of piety and spirituality, even though there had already been examples of this in previous centuries, for example, the founding of monasteries, churches, chapels, hospitals, recollections or assistance to the poor. In line with general European trends, during the 15th–18th centuries, private and personal devotion in Portugal embraced the use of devotional objects, images and books. In this sense, it would be worth rethinking the concept of "devotion" or "devotional practices". Miri Rubin, in her book *Emotion and devotion. The meaning of Mary in Medieval Religious Cultures* (Rubin 2009), or the volume coordinated by Marco Faini and Alessia Meneghin, *Domestic Devotions in Early Modern World* (Faini and Meneghin 2018), emphasized its dynamic nature.

Another dimension that would be worth studying in more depth relates to the "living saints" and their role and influence in the spiritual, social and cultural framework of modern Portugal. The recent publication of issue 33 of the journal *Archivio Italiano per la Storia della pietà,* in honour of Gabriella Zarri[2] (Romagnoli 2020), especially her research focused on the notion of "living saints", inaugurated with the work *Le sante vive,* has rightly drawn attention to its importance. This study had a very significant influence on the framework of research into the history of spirituality, more specifically the dimension related to holiness, seen as a dynamic process conditioned by the coordinates of time and space. In this way, it would be worth studying the universe shaped by the "living saints" in Portugal throughout the 11th–18th centuries, as testimony to a historical and cultural phenomenon that is reflected in the religious, social and even political contexts. In Portugal, as in other Catholic territories, it is possible to trace the cases of various "living saints" who, due to the exceptional nature of their virtues and prophetic and visionary abilities, managed to gather around them groups of devotees who formed communities that contributed to the creation of dynamic relationships around these figures. Hagiography, devotional biography and religious historiography are textual typologies that provide various pieces of information that allow us to reconstitute many of these communities of devotees. In this sense, it seems pertinent that, within the theoretical framework of the history of spirituality, we reconstruct this universe, identifying the people who made up this "devout world" that marked early modern Portugal. On the other hand, listing the relics mentioned in these works will be fundamental to understanding the ways in which networks of devotees were created, which proved to be fundamental in solidifying these "cults".

Despite some contributions in recent years, Portuguese monastic and convent libraries are still an understudied area. It is worth emphasising how monasteries played a central role in the framing of culture and the transmission of knowledge, configuring themselves as spaces/repositories of written memory. The transcription, analysis and identification of the works contained in inventories, catalogues and other documentation will be an essential task in order to determine the extent to which male and female religious houses were centres for the dissemination of knowledge that obeyed different paradigms of knowledge according to gender.

All these dimensions have yet to be studied. While it is true that recent contributions have opened various perspectives, there are still many challenges and several possible research paths in this area of knowledge, which is still rather opaque and could perhaps become clearer as other documentation and sources allow sources to be compared.

**Funding:** This paper was financed by National Funds through the FCT - Foundation for Science and 338 Technology, under the project UIDB/04059/2020.

**Institutional Review Board Statement:** Not applicable.

**Informed Consent Statement:** Not applicable.

**Data Availability Statement:** Data are contained within the article.

**Acknowledgments:** I thank Zulmira Santos for the careful revision of the manuscript.

**Conflicts of Interest:** The author declares no conflict of interest.

## Notes

[1] Álvaro Huerga (1923–2018) was a Spanish theologian and historian, who stood out in the study of the history of spirituality in the 16th century. He is the author of several works, including *Historia de los Alumbrados* (4 volumes, (Huerga 1978).

[2] Gabriella Zarri is a professor of Modern History. Her research has focused on the study of models of holiness, the practice of spiritual direction in Europe in the 16th–18th centuries and women's writing in monastic and conventual spaces.

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
