# Peer review of "Perspectives on Religious History in Early Modern Portugal: Problems, Historiographic Production and Challenges"

_religions, doi:10.3390/rel14111421_

Round 1
Reviewer 1 Report
Comments and Suggestions for Authors
The article can be published.
The main question addressed by the article is an assessment of the paths taken by research, in the last 15 years, in the field of the history of spirituality of the Modern Period in Portugal.
The topic is relevant in the field, as it highlights the most important contributions in this scientific area, as well as the challenges that continue to exist and stimulate the debate in this scientific area. The article complements the perspectives published in História Religiosa de Portugal (3 vols., 2000-2002) and in Zulmira Santos’ article «A produção historiográfica portuguesa sobre a história religiosa na Época Moderna: questões e perspectivas», published in Lusitania Sacra (2009). The research highlights
recent contributions have opened new perspectives in research paths in this area of knowledge.
The methodology is appropriate
The conclusions are consistent with the arguments presented, drawing attention to themes whose study continues to constitute a path to be explored
Author Response
Thank you for your comments. Please find the revised version attached.

Reviewer 2 Report
Comments and Suggestions for Authors
Better define the chronological framework: The short abstract talks about changes in the last decade, but the article – since 2009 (53), then actually starting with the 1950s. Why?
Better define the topic: Spirituality (what is it?) , as in the title, or spirituality AND the history of religion, as argued in the first Par? Or religious, ecclesiastical and spiritual as in line 51?
line 84 – perspective of bookshops but then organization of libraries in monasteries. Mistranslation of biblioteca? (also 356)
Only on line 153, the author moves to what is usually referred to as Spirituality, and then, immediately, to female spirituality. Admittedly, female spirituality is important in early modern Catholicism, but nonetheless the author needs to explain why female spirituality and not first giving a broader overview of forms of spirituality.
Line 170 – “equating the concepts of gender and interiority”: who is doing the equating? And what does equating mean?
Line 203 – unexplained transition back to female spirituality, and 280 back to female aristocratic spirituality.
286 – Bible. It seems to me that this text, defined by the author as a macro-text (what does in mean?) should open the entire discussion. Is there Christian culture that is not based on it?
Comments on the Quality of English LanguageA major language editing is mandatory.
Author Response
Thank you for your comments. Please find the revised version attached

Reviewer 3 Report
Comments and Suggestions for Authors
This study offers a bibliographic survey of recent research into religious history in Portugal. Given that this is a subject little known outside Portugal, it is certainly helpful to have such a survey. By its nature, such a survey does not claim originality in itself. Certainly it fits into a broader trend not unique to Portugal in illustrating a broad shift away from Church history to religious history. I did notice that there was no mention of Jews or Jewish spirituality; I am no specialist here, but I was uncertain whether they were simply absent in the early modern period (which I find hard to believe). The other perspective barely touched on was the relationship between religion and colonialism. I appreciate that the focus was on the region of Portugal rather than on its colonies, but surely colonialism had an influence in shaping religious life in Portugal itself?
The English expression was grammatically correct, but at times it was rather stilted. One minor term, was reference to a study of bookshops run by religious orders. I was curious whether this might refer to libraries rather than bookshops.
Other comments
Abstract
This paper offers a panoramic outline…
The focus of this article is on Portugal within Europe rather than on its colonial spaces in Latin America, Africa and Asia. (also line 54—this looks like a sentence added quickly to the asbstract and the paper
Comments on the Quality of English Language
line 26 the Portuguese panorama ] panorama is strange here. structuring element of political, social and cultural life in Portugal.
142 The secular clergy has also received
239 The polarised issue] is this the right word? Perhaps Contested issues surrounding
242 polarised by] contested by Erasmus etc
336 polarised around] centred/focused around the notion of living saints
348 that allow us to reconstitute]
355 bookshops] does this mean libraries (bookshops are where books are sold)
364 OMIT it cannot be denied that
366 data] sources
Author Response

(The authors gave the same response as above.)

Round 2
Reviewer 2 Report
Comments and Suggestions for Authors
The revisions satisfy.
Comments on the Quality of English LanguageNo comments
Author Response
Thank you for your comments